# Curvature Spinors in Locally Inertial Frame and the Relations with Sedenion

**In Ki Hong [1], Choong Sun Kim [1,2,\*] and Gyung Hyun Min[1]**

[1] Department of Physics and IPAP, Yonsei University, Seoul 03722, Korea; hijko3@yonsei.ac.kr (I.K.H.); mk9538@yonsei.ac.kr (G.H.M.)

[2] Institute of High Energy Physics, Dongshin University, Naju 58245, Korea

\* Correspondence: cskim@yonsei.ac.kr

**Abstract:** In the 2-spinor formalism, the gravity can be dealt with curvature spinors with four spinor indices. Here we show a new effective method to express the components of curvature spinors in the rank-2 $4 \times 4$ tensor representation for the gravity in a locally inertial frame. In the process we have developed a few manipulating techniques, through which the roles of each component of Riemann curvature tensor are revealed. We define a new algebra 'sedon', the structure of which is almost the same as sedenion except for the basis multiplication rule. Finally we also show that curvature spinors can be represented in the sedon form and observe the chiral structure in curvature spinors. A few applications of the sedon representation, which includes the quaternion form of differential Binanchi identity and hand-in-hand couplings of curvature spinors, are also presented.

**Keywords:** spinor formalism; general relativity; sedenion; quaternion; representation theory

## 1. Introduction

In the 2-spinor formalism [1–3] all tensors with spacetime indices can be transformed into spinors with twice the number of spinor indices, i.e., a rank-2 tensor is changed into a spinor with four spinor indices. In addition, if the tensor is antisymmetric and real, it can be represented by a sum of two spinors with two spinor indices, and they are complex conjugate of each other, which indicates that a rank-2 antisymmetric tensor is equivalent to a spinor with two spinor indices. The Riemann curvature tensor is a rank-4 real tensor which describes gravitational fields and it has two antisymmetric characters. This means that the gravity can be described by two spinors with four spinor indices. Those two spinors are called curvature spinors: One of them is Ricci spinor and the other is Weyl conformal spinor [1,3–5].

At any points on a pseudo-Riemannian manifold, we can find a locally flat coordinate [6], whose metric is Minkowski. While the metric is locally Minkowski, the second derivative of the metric is not necessarily zero and the Riemann curvature tensor as well as curvature spinors do not have to be zero. Here we can obtain the explicit representations of curvature spinors, the components of which can be easily identified by using new techniques, i.e., manipulating spinor indices and rotating sigma basis in locally flat coordinates [7]. Then all the components of curvature spinors are represented with simple combinations of Riemann curvature tensors. Here the representations are not described by the four-dimensional basis but by the three-dimensional basis $\otimes$ three-dimensional basis, and thus it suggests a different interpretation of time. The process has been applied on both Ricci spinor and Weyl spinor, which are curvature spinors, and each spinor is described as the sum of two newly defined parts; one of which is a real part and the other is a pure imaginary part. The obtained representation can be used not only in a special flat coordinate but also for vielbein indices or in any other normal coordinates, like Riemann normal coordinate and Fermi coordinate [8–16]. By comparing

the final forms of Ricci spinor with the spinor form of Einstein equation, we figure out the roles of each component of Riemann curvature tensor, whose components serve as momentum, energy or stress of gravitational fields. Furthermore, we show that the components of Weyl conformal spinor can be represented as a simple combination of Wely tensors in flat coordinate.

There are already quite a few papers that show the relation between gravitational fields and Cayley–Dickson algebras including sedenion; however, all papers are restricted to a weak gravitational field in a flat frame [17–22]. Here we express the basis of sedenion as a set of direct products of a quaternion basis, through which we can define a new algebra 'sedon', whose structure is similar to sedenion except for the basis multiplication rule. We show that the curvature spinors for general gravitational fields in locally flat coordinates can be regarded as a sedon. The spinors are described on the direct procduct of totally seperated left-handed basis and right-handed quaternion basis. From this, we can get a view of the gravitational effects as the combination of right-handed and left-handed rotational effects. We also introduce a few applications of the sedon form with multiplication techniques. One of the application is the quaternion form of differential Bianchi identity and, in the process, we introduce a new index notation with the spatially opposite-handed quantities.

## 2. Tensor Representation of a Field with Two Spinor Indices

In this section we introduce the basics about the 2-spinor formalism, which have been already explained in detail in our earlier paper [7]. We use the front part of Latin small letters $a, b, ..., h$ and Greek letters $\mu, \nu, \rho...$ as four dimensional space-time indices, which can be $0, 1, 2$ or $3$. The later part of Latin small letters $i, j, ...,$ which can be $1, 2$ or $3$, are used as three dimensional indices.

Any tensor $T_{abc..}$ with spacetime indices $a, b, c, ..,$ can be inverted into a spinor with spinor indices $A, A', B, B', ..$ like $T_{AA'BB'..}$ by multiplying Infeld-van der Waerden symbols $g_{AA'}{}^{a}$,

$$T_{AA'BB'..} = T_{ab..} g_{AA'}{}^{a} g_{BB'}{}^{a} .. \quad . \tag{1}$$

In Minkowski spacetime, $g_{AA'}{}^{a}$ is $\frac{1}{\sqrt{2}} \sigma^{a}{}_{AA'}$, where $\sigma^{a}{}_{AA'}$ are four-sigma matrices $(\sigma^0, \sigma^1, \sigma^2, \sigma^3)$; $\sigma^0$ is $2 \times 2$ identity matrix and $\sigma^1, \sigma^2, \sigma^3$ are Pauli matrices. Equation (1) can be written conventionally as

$$T_{AA'BB'..} = T_{ab..}. \tag{2}$$

Any arbitrary anti-symmetric tensor $F_{ab} = F_{AA'BB'}$ can be expressed as the sum of two symmetric spinors as

$$F_{AA'BB'} = \varphi_{AB} \varepsilon_{A'B'} + \varepsilon_{AB} \psi_{A'B'}, \tag{3}$$

where $\varphi_{AB} = \frac{1}{2} F_{ABC'}{}^{C'}$ and $\psi_{A'B'} = \frac{1}{2} F_{C}{}^{C}{}_{A'B'}$ are symmetric spinors (unprimed and primed spinor indices can be switched back and forth each other), and $\varepsilon^{AB}, \varepsilon^{A'B'}, \varepsilon_{AB}, \varepsilon_{A'B'}$ are the $\varepsilon$-spinors whose components are $\varepsilon^{12} = \varepsilon_{12} = +1, \varepsilon^{21} = \varepsilon_{21} = -1$ [1]; $k^A = \varepsilon^{AB} k_B, k_B = k^A \varepsilon_{AB}$. If $F_{ab}$ is real, then $\psi_{A'B'} = \bar{\varphi}_{A'B'}$ (where $\bar{\varphi}$ is the complex conjugate of $\varphi$) and

$$F_{ab} = F_{AA'BB'} = \varphi_{AB} \varepsilon_{A'B'} + \varepsilon_{AB} \bar{\varphi}_{A'B'}. \tag{4}$$

We have shown the components of $\varphi_{AB}$ and $\bar{\varphi}_{A'B'}$ explicitly in flat spacetime in [7]. The sign conventions for the Minkowski metric is $g_{\mu\nu} = \text{diag}(1, -1, -1, -1)$.

For any real anti-symmetric tensor $F_{AA'BB'}$, we can write as

$$F_{AA'BB'} = \frac{1}{2} F_{\mu\nu} \sigma^{\mu}{}_{AA'} \sigma^{\nu}{}_{BB'} = \frac{1}{2} F_{\mu\nu} \sigma^{\mu}{}_{AA'} \bar{\sigma}^{\nu \, C'C} \varepsilon_{C'B'} \varepsilon_{CB}, \tag{5}$$

where $\bar{\sigma}^\mu = (\sigma^0, -\sigma^1, -\sigma^2, -\sigma^3)$, then

$$
\begin{aligned}
\varphi_{AB} &= \frac{1}{2}F_{AA'B}{}^{A'} = \frac{1}{2}F_{AA'BB'}\varepsilon^{A'B'} \\
&= \frac{1}{4}F_{\mu\nu}\sigma^\mu_{AA'}\bar{\sigma}^{\nu\,C'C}\varepsilon_{C'B'}\varepsilon_{CB}\varepsilon^{A'B'} = \frac{1}{4}F_{\mu\nu}\,\sigma^\mu_{AA'}\bar{\sigma}^{\nu\,A'C}\varepsilon_{CB}\,,
\end{aligned}
\tag{6}
$$

$$
\begin{aligned}
\bar{\varphi}_{A'B'} &= \frac{1}{2}F_{AA'}{}^A{}_{B'} = \frac{1}{2}F_{AA'BB'}\varepsilon^{AB} \\
&= \frac{1}{4}F_{\mu\nu}\bar{\sigma}^{\mu C'C}\sigma^\nu_{BB'}\varepsilon_{A'C'}\varepsilon_{AC}\varepsilon^{AB} = \frac{1}{4}F_{\mu\nu}\,\varepsilon_{A'C'}\bar{\sigma}^{\mu C'B}\sigma^\nu_{BB'}\,,
\end{aligned}
\tag{7}
$$

where we used the relation $g_{BB'}{}^\nu = \frac{1}{\sqrt{2}}\sigma^\mu_{BB'} = \frac{1}{\sqrt{2}}\bar{\sigma}^{\nu\,C'C}\varepsilon_{C'B'}\varepsilon_{CB}$. This can be established when the space-time metric is locally flat [1,23]. If we apply the relations to the general coordinates with relevant modifications, the results for arbitrary coordinates can be obtained.

Since

$$
\sigma^\mu_{AA'}\bar{\sigma}^{\nu\,A'C} =
\begin{pmatrix}
\sigma^0\sigma^0 & -\sigma^0\sigma^1 & -\sigma^0\sigma^2 & -\sigma^0\sigma^3 \\
\sigma^1\sigma^0 & -\sigma^1\sigma^1 & -\sigma^1\sigma^2 & -\sigma^1\sigma^3 \\
\sigma^2\sigma^0 & -\sigma^2\sigma^1 & -\sigma^2\sigma^2 & -\sigma^2\sigma^3 \\
\sigma^3\sigma^0 & -\sigma^3\sigma^1 & -\sigma^3\sigma^2 & -\sigma^3\sigma^3
\end{pmatrix}_A{}^C
=
\begin{pmatrix}
\sigma^0 & -\sigma^1 & -\sigma^2 & -\sigma^3 \\
\sigma^1 & -\sigma^0 & -i\sigma^3 & i\sigma^2 \\
\sigma^2 & i\sigma^3 & -\sigma^0 & -i\sigma^1 \\
\sigma^3 & -i\sigma^2 & i\sigma^1 & -\sigma^0
\end{pmatrix}_A{}^C\,,
\tag{8}
$$

$\varphi_A{}^D = \varepsilon^{DB}\varphi_{AB}$ becomes

$$
\begin{aligned}
\varphi_A{}^D &= \frac{1}{4}F_{\mu\nu}\sigma^\mu_{AA'}\bar{\sigma}^{\nu\,A'D} \\
&= \frac{1}{4}\left[
\begin{pmatrix}
0 & -F_{10} & -F_{20} & -F_{30} \\
F_{10} & 0 & F_{12} & F_{13} \\
F_{20} & -F_{12} & 0 & F_{23} \\
F_{30} & -F_{13} & -F_{23} & 0
\end{pmatrix}
\begin{pmatrix}
\sigma^0 & -\sigma^1 & -\sigma^2 & -\sigma^3 \\
\sigma^1 & -\sigma^0 & -i\sigma^3 & i\sigma^2 \\
\sigma^2 & i\sigma^3 & -\sigma^0 & -i\sigma^1 \\
\sigma^3 & -i\sigma^2 & i\sigma^1 & -\sigma^0
\end{pmatrix}^T
\right]_A{}^D \\
&= \frac{1}{2}\left(F_{i0}\sigma^i - \frac{1}{2}i\,\epsilon^{ij}{}_k F_{ij}\sigma^k\right)_A{}^D\,,
\end{aligned}
\tag{9}
$$

where $i$, $j$, $k$ are the three-dimensional vector indices which have the value 1, 2 or 3, and $\epsilon^{ij}{}_k$ is $\epsilon_{pqk}\delta^i_p\delta^j_q$ for the Levi–Civita symbol $\epsilon_{ijk}$. Einstein summation convention is used for three-dimensional vector indices $i, j$ and $k$. Similar to Equations (8) and (9),

$$
\bar{\sigma}^{\mu C'B}\sigma^\nu_{BB'} =
\begin{pmatrix}
\sigma^0\sigma^0 & \sigma^0\sigma^1 & \sigma^0\sigma^2 & \sigma^0\sigma^3 \\
-\sigma^1\sigma^0 & -\sigma^1\sigma^1 & -\sigma^1\sigma^2 & -\sigma^1\sigma^3 \\
-\sigma^2\sigma^0 & -\sigma^2\sigma^1 & -\sigma^2\sigma^2 & -\sigma^2\sigma^3 \\
-\sigma^3\sigma^0 & -\sigma^3\sigma^1 & -\sigma^3\sigma^2 & -\sigma^3\sigma^3
\end{pmatrix}^{C'}{}_{B'}
=
\begin{pmatrix}
\sigma^0 & \sigma^1 & \sigma^2 & \sigma^3 \\
-\sigma^1 & -\sigma^0 & -i\sigma^3 & i\sigma^2 \\
-\sigma^2 & i\sigma^3 & -\sigma^0 & -i\sigma^1 \\
-\sigma^3 & -i\sigma^2 & i\sigma^1 & -\sigma^0
\end{pmatrix}^{C'}{}_{B'}\,,
\tag{10}
$$

$$
\bar{\varphi}^{D'}{}_{B'} = \varepsilon^{D'A'}\bar{\varphi}_{A'B'} = -\frac{1}{4}F_{\mu\nu}\,\bar{\sigma}^{\mu D'B}\sigma^\nu_{BB'} = \frac{1}{2}\left(F_{i0}\sigma^i + \frac{1}{2}i\,\epsilon^{ij}{}_k F_{ij}\sigma^k\right)^{D'}{}_{B'}.
\tag{11}
$$

If we denote matrix representation of $\varepsilon_{AB}$ by $\varepsilon$, then

$$
\sigma^\mu\varepsilon = (\sigma^0, \sigma^1, \sigma^2, \sigma^3)\varepsilon = (i\sigma^2, -\sigma^3, i\sigma^0, \sigma^1),
\tag{12}
$$

$$
\varepsilon\sigma^\mu = \varepsilon(\sigma^0, \sigma^1, \sigma^2, \sigma^3) = (i\sigma^2, \sigma^3, i\sigma^0, -\sigma^1).
\tag{13}
$$

Let us define $s^\mu$ and $\bar{s}^\mu$ as

$$
s^0 = i\sigma^2, \quad s^1 = -\sigma^3, \quad s^2 = i\sigma^0, \quad s^3 = \sigma^1,
\tag{14}
$$

$$\bar{s}^0 = i\sigma^2, \quad \bar{s}^1 = -\sigma^3, \quad \bar{s}^2 = -i\sigma^0, \quad \bar{s}^3 = \sigma^1, \tag{15}$$

where $\bar{s}^\mu$ is complex conjugate of $s^\mu$. Then

$$\sigma^\mu \varepsilon = (s^0, s^1, s^2, s^3), \tag{16}$$

$$\varepsilon \sigma^\mu = (s^0, -s^1, s^2, -s^3) = (\bar{s}^0, -\bar{s}^1, -\bar{s}^2, -\bar{s}^3), \tag{17}$$

and

$$\varphi_{AB} \quad = \varphi_A{}^D \varepsilon_{DB} = \frac{1}{2}(F_{i0}s^i - \frac{1}{2}i\,\epsilon^{ij}{}_k F_{ij}s^k), \tag{18}$$

$$\varphi_{A'B'} \quad = \varphi^{D'}{}_{B'}\varepsilon_{D'A'} = -\varepsilon_{A'D'}\phi^{D'}{}_{B'} = \frac{1}{2}(F_{i0}\bar{s}^i + \frac{1}{2}i\,\epsilon^{ij}{}_k F_{ij}\bar{s}^k), \tag{19}$$

where $s^i$ have unprimed indices $s^i = s^i_{AB}$ and $\bar{s}^i$ have primed indices $\bar{s}^i = \bar{s}^i_{A'B'}$.

## 3. Einstein Field Equations and Curvature Spinors

In this section we introduce the basics about general relativity in the 2-spinor formalism, and flat coordinates on the pseudo-Riemannian manifold. For a (torsion-free) Riemann curvature tensor

$$R^\mu{}_{\nu\rho\sigma} = \partial_\rho \Gamma^\mu{}_{\nu\sigma} - \partial_\sigma \Gamma^\mu{}_{\nu\rho} + \Gamma^\lambda{}_{\nu\sigma}\Gamma^\mu{}_{\lambda\rho} - \Gamma^\lambda{}_{\nu\rho}\Gamma^\mu{}_{\lambda\sigma}, \tag{20}$$

where $\Gamma^\rho{}_{\mu\nu}$ is a Chistoffel symbol

$$\Gamma^\rho{}_{\mu\nu} = \frac{1}{2}g^{\rho\lambda}(\partial_\mu g_{\nu\lambda} + \partial_\nu g_{\mu\lambda} - \partial_\lambda g_{\mu\nu}). \tag{21}$$

Here $R_{\mu\nu\rho\sigma}$ has follwing properties [6]

$$R_{\mu\nu\rho\sigma} = -R_{\nu\mu\rho\sigma}, \tag{22}$$

$$R_{\mu\nu\rho\sigma} = -R_{\mu\nu\sigma\rho}, \tag{23}$$

$$R_{\mu\nu\rho\sigma} = R_{\rho\sigma\mu\nu}. \tag{24}$$

In short, we can denote as

$$R_{\mu\nu\rho\sigma} = R_{([\mu\nu][\rho\sigma])}, \tag{25}$$

where parentheses ( ) and square brackets [ ] indicates symmetrization and anti-symmetrization of the indices [6]. The Riemann curvature tensor has two kinds of Bianchi identities

$$R_{\mu[\nu\rho\sigma]} = 0, \tag{26}$$

$$\nabla_{[\lambda}R_{\mu\nu]\rho\sigma} = 0, \tag{27}$$

where $\nabla_\lambda A^\mu = \partial_\lambda A^\mu + \Gamma^\mu{}_{\nu\lambda}A^\nu$.

From the antisymmetric properties of Riemann curvature tensor, it can be decomposed into sum of curvature spinors, $X_{ABCD}$ and $\Phi_{ABC'D'}$, as

$$R_{abcd} \quad = \frac{1}{2}R_{AX'B}{}^{X'}{}_{cd}\varepsilon_{A'B'} + \frac{1}{2}R_{XA}{}^X{}_{B'}{}_{cd}\varepsilon_{AB}$$
$$= \Phi_{ABC'D'}\epsilon_{A'B'}\epsilon_{CD} + \Phi_{A'B'CD}\epsilon_{AB}\epsilon_{C'D'} + X_{ABCD}\epsilon_{A'B'}\epsilon_{C'D'} + \bar{X}_{A'B'C'D'}\epsilon_{AB}\epsilon_{CD}, \tag{28}$$

where

$$X_{ABCD} = R_{AX'B}{}^{X'}{}_{CY'D}{}^{Y'}, \qquad \Phi_{ABC'D'} = R_{AX'B}{}^{X'}{}_{YC'}{}^{Y}{}_{D'}. \tag{29}$$

The totally symmetric part of $X_{ABCD}$

$$\Psi_{ABCD} = X_{A(BCD)} = X_{(ABCD)} \tag{30}$$

is called gravitational spinor or Weyl conformal spinor, and $\Phi_{ABC'D'}$ is referred as Ricci spinor [1,4,5]. It is well known that

$$\Phi_{AA'BB'} = \Phi_{ab} = \Phi_{ba} = \bar{\Phi}_{ab}, \qquad \Phi_a{}^a = 0, \tag{31}$$

and Einstein tensor is

$$G_{ab} = R_{ab} - \frac{1}{2}Rg_{ab} = -\Lambda g_{ab} - 2\Phi_{ab}, \tag{32}$$

where $\Lambda = X_{AB}{}^{AB}$, which is equal to $R/4$ [1]. Therefore, the Einstein field equation

$$G_{ab} + \lambda g_{ab} = 8\pi G T_{ab}, \tag{33}$$

where $\lambda$ is a cosmology constant, can be written in the form

$$\Phi_{ab} = 4\pi G(-T_{ab} + \frac{1}{4}T_q^q g_{ab}), \qquad \Lambda = -2\pi G T_q^q + \lambda. \tag{34}$$

Since any symmetric tensor $U_{ab}$ can be expressed as

$$U_{ab} = U_{AA'BB'} = S_{ABA'B'} + \varepsilon_{AB}\varepsilon_{A'B'}\tau, \tag{35}$$

where $\tau = \frac{1}{4}T_c^c$ and $S_{ABA'B'}$ is traceless and symmetric [1], the traceless part of the energy-momentum (symmetric) tensor $T_{ab}$ can be written by $S_{ab} = T_{ab} - \frac{1}{4}T_c^c g_{ab}$. Therefore, the spinor form of Einstein Equations (34) becomes

$$\Phi_{ABA'B'} = -4\pi G S_{ABA'B'}, \qquad X_{AB}{}^{AB} = -8\pi G\tau + \lambda. \tag{36}$$

Weyl tensor $C_{\mu\nu\rho\sigma}$ which is another measure of the curvature of spacetime, like Riemann curvature tensor, is defined as [6,24]

$$C_{\mu\nu\rho\sigma} = R_{\mu\nu\rho\sigma} + \frac{1}{2}(R_{\mu\sigma}g_{\nu\rho} - R_{\mu\rho}g_{\nu\sigma} + R_{\nu\rho}g_{\mu\sigma} - R_{\nu\sigma}g_{\mu\sigma}) + \frac{1}{6}R(g_{\mu\rho}g_{\sigma\nu} - g_{\mu\sigma}g_{\nu\rho}). \tag{37}$$

It has the same propterties as Equations (22), (23) and (26). It is known [1] that Weyl tensor has the following relationship with Weyl conformal spinor $\Psi_{ABCD}$:

$$C_{abcd} = \Psi_{ABCD}\varepsilon_{A'B'}\varepsilon_{C'D'} + \bar{\Psi}_{A'B'C'D'}\varepsilon_{AB}\varepsilon_{CD}. \tag{38}$$

At any point $P$ on the pseudo-Riemannian manifold, we can find a flat coordinate system, such that,

$$g_{\mu\nu}(P) = \eta_{\mu\nu}, \qquad \left.\frac{\partial g_{\mu\nu}}{\partial x^\lambda}\right|_P = 0, \tag{39}$$

where $g_{\mu\nu}(P)$ is the metric at the point $P$ and $\eta_{\mu\nu}$ is the Minkowski metric. In this coordinate system, while the Christoffel symbol is zero, the Riemann curvature tensor is [25,26]

$$R_{\mu\nu\rho\sigma} = \frac{1}{2}(\partial_\nu\partial_\rho g_{\mu\sigma} - \partial_\nu\partial_\sigma g_{\mu\rho} + \partial_\mu\partial_\sigma g_{\nu\rho} - \partial_\mu\partial_\rho g_{\nu\sigma}). \tag{40}$$

For future use we introduce Fermi coordinate, which is one of the locally flat coordinate whose time axis is a tangent of a geodesic. The coordinate follows the Fermi conditions

$$g_{\mu\nu}|_G = \eta_{\mu\nu}, \qquad \Gamma^\rho_{\mu\nu}|_G = 0, \tag{41}$$

along the geodesic G.

## 4. The Tensor Representation of Curvature Spinors

In this section we show the process of representing curvature spinors in $4 \times 4$ matrices or $3 \times 3$ matrices. We discuss physical implications of those representations. From now on, we will always use locally flat coordinate for all spacetime indices.

From Equations (4) and (28), we can lead to

$$\begin{aligned}
R_{abcd} &= \phi_{AB,cd}\varepsilon_{A'B'} + \varepsilon_{AB}\bar{\phi}_{A'B',cd} \\
&= \Phi_{ABC'D'}\epsilon_{A'B'}\epsilon_{CD} + \Phi_{A'B'CD}\epsilon_{AB}\epsilon_{C'D'} + X_{ABCD}\epsilon_{A'B'}\epsilon_{C'D'} + \bar{X}_{A'B'C'D'}\epsilon_{AB}\epsilon_{CD},
\end{aligned} \tag{42}$$

where

$$\phi_{AB,cd} = \frac{1}{2}(R_{i0\ cd}s^i - \frac{1}{2}i\,\epsilon_{ijk}R_{ij\ cd}s^k), \tag{43}$$

$$\bar{\phi}_{A'B',cd} = \frac{1}{2}(R_{i0\ cd}\bar{s}^i + \frac{1}{2}i\,\epsilon_{ijk}R_{ij\ cd}\bar{s}^k), \tag{44}$$

from Equations (18) and (19). We write here the form of $\epsilon^{ij}{}_k$ as $\epsilon_{ijk}$ for convenience; it is not so difficult to recover the upper- and lower-indices. By decomposing $\phi_{AB,cd}$ one more times, we get

$$\begin{aligned}
\Phi_{ABC'D'} &= \frac{1}{4}(R_{i0\ j0}s^i\bar{s}^j + \frac{1}{2}i\epsilon_{pqr}R_{i0\ pq}s^i\bar{s}^r - \frac{1}{2}i\epsilon_{ijk}R_{ij\ l0}s^k\bar{s}^l + \frac{1}{4}\epsilon_{ijk}\epsilon_{pqr}R_{ij\ pq}s^k\bar{s}^r), \\
&= \frac{1}{4}(R_{k0\ l0} + \frac{1}{2}i\epsilon_{pql}R_{k0\ pq} - \frac{1}{2}i\epsilon_{ijk}R_{ij\ l0} + \frac{1}{4}\epsilon_{ijk}\epsilon_{pql}R_{ij\ pq})s^k\bar{s}^l,
\end{aligned} \tag{45}$$

$$\begin{aligned}
X_{ABCD} &= \frac{1}{4}(R_{0i\ 0j}s^is^j - \frac{1}{2}i\epsilon_{pqr}R_{0i\ pq}s^is^r - \frac{1}{2}i\epsilon_{ijk}R_{ij\ 0l}s^ks^l - \frac{1}{4}\epsilon_{ijk}\epsilon_{pqr}R_{ij\ pq}s^ks^r) \\
&= \frac{1}{4}(R_{0k\ 0l} - \frac{1}{2}i\epsilon_{pql}R_{0k\ pq} - \frac{1}{2}i\epsilon_{ijk}R_{ij\ 0l} - \frac{1}{4}\epsilon_{ijk}\epsilon_{pql}R_{ij\ pq})s^ks^l.
\end{aligned} \tag{46}$$

We note that $\Phi$ and $X$ are expressed with two three-dimensional basis like the form in $3 \times 3$ basis. Even though there is no 0-th base, which may be related to the curvature of time, $\Phi$ and $X$ can fully describe the spacetime structure. If $\Phi$ and $X$ are represented in Fermi coordinate, the disappearance of 0-th compomonents of the $s^i$ basis may come from the fact that time follows proper time. However, since here Equations (45) and (46) are expressed not only in Fermi coordinate but also in general locally flat coordinates, whose 0-th coordinate may not be time direction, the representations like Equations (45) and (46) may demand a new interpretation of the curvature of time, which is not just as a component of fourth (or 0-th) axis in 4-dimensional space-time. Technically the interpretation of the space-time structure, which was considered to be a bundle of four directions, might be reconsidered.

We can divide Equation (45) into two terms by defining

$$P_{ij} \equiv \frac{1}{2}\epsilon_{pqj}R_{i0\ pq} - \frac{1}{2}\epsilon_{pqi}R_{j0\ pq}, \tag{47}$$

$$S_{ij} \equiv R_{0i\ 0j} + \frac{1}{4}\epsilon_{pqi}\epsilon_{rsj}R_{pq\ rs}, \tag{48}$$

$$\Theta_{ij} \equiv R_{i0\ j0} + \frac{1}{2}i\epsilon_{pqj}R_{i0\ pq} - \frac{1}{2}i\epsilon_{pqi}R_{pq\ j0} + \frac{1}{4}\epsilon_{pqi}\epsilon_{rsj}R_{pq\ rs} = S_{ij} + i\,P_{ij}, \tag{49}$$

where $P_{ij}$ is anti-symmetric and $S_{ij}$ is symmetric for $i, j$. Then Equation (45) is represented as

$$\Phi_{ABC'D'} = \frac{1}{4}\Theta_{ij}s^{i}_{AB}\bar{s}^{j}_{C'D'}. \tag{50}$$

The components of $P_{ij}$ and $S_{ij}$ can be simply expressed as

$$-\frac{1}{2}\epsilon_{ijk}P_{ij} \quad = -\frac{1}{4}(\epsilon_{ijk}\epsilon_{pqj}R_{i0\ pq} - \epsilon_{ijk}\epsilon_{pqi}R_{j0\ pq}) = R_{0i\ ki}, \tag{51}$$

$$S_{\underline{i}\underline{j}} \quad = R_{0\underline{i}\ 0\underline{j}} + \varepsilon_{\underline{i}\ pq}\varepsilon_{\underline{j}\ rs}R_{pq\ rs}, \tag{52}$$

$$S_{\underline{i}\underline{i}} \quad = R_{0\underline{i}\ 0\underline{i}} + |\varepsilon_{\underline{i}\ pq}|R_{pq\ pq}, \tag{53}$$

where the underlined symbols in subscripts are the value-fixed indices which does not sum up for dummy indices; one of example is $S_{11} = R_{01\ 01} + R_{23\ 23}$.

We can express $\Phi_{ABCD}$ as a tensor by multiplying $g_{\mu}^{AC'}$, which is

$$g_{\mu}^{AB'} = \varepsilon^{AC}\varepsilon^{B'D'}g_{\mu\nu}g^{\nu}_{CD'} = \frac{1}{\sqrt{2}}(\sigma^0, \sigma^1, -\sigma^2, \sigma^3)^{AB'} = \frac{1}{\sqrt{2}}\sigma^{t\ AB'}_{\mu} = \frac{1}{\sqrt{2}}\sigma^{*\ AB'}_{\mu}, \tag{54}$$

where sigma matrices with superscript $\sigma^t$ and $\sigma^*$ mean the transpose and the complex conjugate of $\sigma$. To calculate $\Phi_{ABC'D'}g_{\mu}^{AC'}g_{\nu}^{BD'} = (1/4\,\Theta_{ij}s^{i}_{AB}\bar{s}^{j}_{C'D'})g_{\mu}^{AC'}g_{\nu}^{BD'}$, let us define

$$f(k,l)_{\mu\nu} = \sigma^{k}_{AB}\sigma^{l}_{C'D'}g_{\mu}^{AC'}g_{\nu}^{BD'} = \frac{1}{2}(\sigma^{k}_{AB}\sigma^{l}_{C'D'})\sigma^{*\ AC'}_{\mu}\sigma^{*\ BD'}_{\nu}. \tag{55}$$

Values of $f(k,l)_{\mu\nu}$ are shown in Table 1.

**Table 1.** The lists of $f(k,l)_{\mu\nu} = \sigma^{k}_{AB}\sigma^{l}_{C'D'}g_{\mu}^{AC'}g_{\nu}^{BD'}$.

| | | |
|---|---|---|
| $f(0,1)_{\mu\nu} = \begin{pmatrix} 0 & 1 & 0 & 0 \\ 1 & 0 & 0 & 0 \\ 0 & 0 & 0 & i \\ 0 & 0 & i & 0 \end{pmatrix},$ | $f(1,0)_{\mu\nu} = \begin{pmatrix} 0 & 1 & 0 & 0 \\ 1 & 0 & 0 & 0 \\ 0 & 0 & 0 & -i \\ 0 & 0 & -i & 0 \end{pmatrix},$ | $f(3,1)_{\mu\nu} = \begin{pmatrix} 0 & 0 & i & 0 \\ 0 & 0 & 0 & 1 \\ i & 0 & 0 & 0 \\ 0 & 1 & 0 & 0 \end{pmatrix},$ |
| $f(1,3)_{\mu\nu} = \begin{pmatrix} 0 & 0 & -i & 0 \\ 0 & 0 & 0 & 1 \\ -i & 0 & 0 & 0 \\ 0 & 1 & 0 & 0 \end{pmatrix},$ | $f(0,3)_{\mu\nu} = \begin{pmatrix} 0 & 0 & 0 & 1 \\ 0 & 0 & -i & 0 \\ 0 & -i & 0 & 0 \\ 1 & 0 & 0 & 0 \end{pmatrix},$ | $f(3,0)_{\mu\nu} = \begin{pmatrix} 0 & 0 & 0 & 1 \\ 0 & 0 & i & 0 \\ 0 & i & 0 & 0 \\ 1 & 0 & 0 & 0 \end{pmatrix},$ |
| $f(0,0)_{\mu\nu} = \begin{pmatrix} 1 & 0 & 0 & 0 \\ 0 & 1 & 0 & 0 \\ 0 & 0 & -1 & 0 \\ 0 & 0 & 0 & 1 \end{pmatrix},$ | $f(1,1)_{\mu\nu} = \begin{pmatrix} 1 & 0 & 0 & 0 \\ 0 & 1 & 0 & 0 \\ 0 & 0 & 1 & 0 \\ 0 & 0 & 0 & -1 \end{pmatrix},$ | $f(3,3)_{\mu\nu} = \begin{pmatrix} 1 & 0 & 0 & 0 \\ 0 & -1 & 0 & 0 \\ 0 & 0 & 1 & 0 \\ 0 & 0 & 0 & 1 \end{pmatrix}.$ |

Using this table, we get $4 \times 4$ representation of $\Phi_{ABC'D'}$ as

$$\Phi_{\mu\nu} = \Phi_{ABC'D'}g_{\mu}^{AC'}g_{\nu}^{BD'}$$

$$
= \frac{1}{4}
\begin{bmatrix}
(\Theta_{12}s^1\bar{s}^2 + \Theta_{21}s^2\bar{s}^1) \\
+(\Theta_{23}s^2\bar{s}^3 + \Theta_{32}s^3\bar{s}^2) \\
+(\Theta_{31}s^3\bar{s}^1 + \Theta_{13}s^1\bar{s}^3) \\
+(\Theta_{11}s^1\bar{s}^1 + \Theta_{22}s^2\bar{s}^2 + \Theta_{33}s^3\bar{s}^3)
\end{bmatrix}_{ABC'D'} g_\mu{}^{AC'}g_\nu{}^{BD'}
$$

$$
= \frac{1}{4}
\begin{bmatrix}
(i\Theta_{12}f(3,0) - i\Theta_{21}f(0,3)) \\
+(i\Theta_{23}f(0,1) - i\Theta_{32}f(1,0)) \\
+(-\Theta_{31}f(1,3) - \Theta_{13}f(3,1)) \\
+(\Theta_{11}f(3,3) + \Theta_{22}f(0,0) + \Theta_{33}f(1,1))
\end{bmatrix}_{\mu\nu}
$$

$$
= \frac{1}{2}
\begin{pmatrix}
\frac{1}{2}(S_{11}+S_{22}+S_{33}) & -P_{23} & -P_{31} & -P_{12} \\
-P_{23} & \frac{1}{2}(-S_{11}+S_{22}+S_{33}) & -S_{12} & -S_{31} \\
-P_{31} & -S_{12} & \frac{1}{2}(S_{11}-S_{22}+S_{33}) & -S_{32} \\
-P_{12} & -S_{31} & -S_{32} & \frac{1}{2}(S_{11}+S_{22}-S_{33})
\end{pmatrix}, \quad (56)
$$

which is a real tensor and $\Phi_{\mu\nu}\eta^{\mu\nu} = 0$, as expected.

From Equations (32), (33) and (56), we can find that $P_{ij}$ and $S_{ij}$ are also non-diagonal components of $G_{\mu\nu}$ and $T_{\mu\nu}$. By comparing Equation (34) with Equation (56), we can interpret $P_{ij}/(8\pi G)$ as a momentum and $S_{ij}/(8\pi G)$ as a stress of a spacetime fluctuation. We can also observe from Equations (47) and (48) that the component of the Riemann curvature tensor of the form $R_{j0\ pq}$ is linked to a momentum, and the form $R_{i0\ j0}$, $R_{pq\ rs}$ linked to a stress-energy.

Now we investigate $X_{ABCD}$ and $\Psi_{ABCD}$ more in detail. Before representing $X$ and $\Psi$ in matrix form, we can check Equation (46) to find out whether $\Lambda = X_{AB}{}^{AB} = R/4$ or not. From the properties of Riemann curvature tensor, Ricci scalar is

$$
R = R_{\mu\nu}{}^{\mu\nu} = 2R_{0i}{}^{0i} + R_{ij}{}^{ij} = 2R_{0i\rho\sigma}g^{\rho 0}g^{\sigma i} + R_{ij\rho\sigma}g^{\rho i}g^{\sigma j}. \tag{57}
$$

For Minckowski metric $g_{\mu\nu} = \eta_{\mu\nu}$, $R$ becomes

$$
R = -2R_{i0i0} + R_{ijij}. \tag{58}
$$

Because

$$
s^k_{AB}s^l_{CD}\varepsilon^{CA}\varepsilon^{DB} = \sigma^k_A{}^P\varepsilon_{PB}\sigma^l_C{}^Q\varepsilon_{QD}\varepsilon^{CA}\varepsilon^{DB} = (\varepsilon^{CA}\sigma^k_A{}^D)(\sigma^l_C{}^Q\varepsilon_{QD})
$$

$$
= -Tr(\bar{s}^k s^l) = \begin{pmatrix} -2 & (k=l) \\ 0 & (k\neq l) \end{pmatrix}, \tag{59}
$$

we can finally see that

$$
\begin{aligned}
X_{AB}{}^{AB} &= X_{ABCD}\varepsilon^{CA}\varepsilon^{DB} \\
&= \frac{1}{4}\left(-2R_{0l\ 0l} + \frac{1}{2}\epsilon_{ijl}\epsilon_{pql}R_{ij\ pq}\right) = \frac{1}{4}(-2R_{0l\ 0l} + R_{ij\ ij}) = \frac{R}{4}
\end{aligned} \tag{60}
$$

from Equation (46). We have used $\epsilon_{pql}R_{0l\ pq} = 0$ by Bianchi identity.

To represent the spinors $X$ and $\Psi$ in simple matrix forms, we first define

$$
Q_{ij} \equiv \frac{1}{2}\epsilon_{pqj}R_{i0\ pq} + \frac{1}{2}\epsilon_{pqi}R_{j0\ pq}, \tag{61}
$$

$$
E_{ij} \equiv R_{0i\ 0j} - \frac{1}{4}\epsilon_{pqi}\epsilon_{rsj}R_{pq\ rs}, \tag{62}
$$

$$
\Xi_{ij} \equiv R_{i0\ j0} - \frac{1}{4}\epsilon_{pqi}\epsilon_{rsj}R_{pq\ rs} - \frac{1}{2}i\epsilon_{pqj}R_{i0\ pq} - \frac{1}{2}i\epsilon_{pqi}R_{pq\ j0} = E_{ij} - iQ_{ij}, \tag{63}
$$

where $Q_{ij}$ and $E_{ij}$ both are symmetric for $i, j$. Then, we have

$$X_{ABCD} = \frac{1}{4}\Xi_{ij}s^i_{AB}s^j_{CD},\tag{64}$$

from Equation (46). This can be expressed in a $4 \times 4$ matrix form by multiplying the factors in a similar way to the Equation (56), but here it is useful to multiply by $(\sigma\varepsilon)^{*\,AC}_\mu(\sigma\varepsilon)^{*\,BD}_\nu = \bar{s}^{AC}_\mu\bar{s}^{BD}_\nu$ for simplicity, instead of $\sigma^{AC}_\mu\sigma^{BD}_\nu$, where the components of $\bar{s}^{AC}_\mu$ is equal to $\bar{s}^\mu$ defined in Equation (15):

$$
\begin{aligned}
X_{ABCD}&\bar{s}^{AC}_\mu\bar{s}^{BD}_\nu = (\frac{1}{4}\Xi_{ij}s^i_{AB}s^j_{CD})\bar{s}^{AC}_\mu\bar{s}^{BD}_\nu \\
&= \frac{1}{2}\begin{pmatrix} -\Xi_{11}-\Xi_{22}-\Xi_{33} & -i\Xi_{23}+i\Xi_{32} & i\Xi_{13}-i\Xi_{31} & -i\Xi_{12}+i\Xi_{21} \\ -i\Xi_{23}+i\Xi_{32} & \Xi_{11}-\Xi_{22}-\Xi_{33} & \Xi_{12}+\Xi_{21} & \Xi_{13}+\Xi_{31} \\ i\Xi_{13}-i\Xi_{31} & \Xi_{12}+\Xi_{21} & -\Xi_{11}+\Xi_{22}-\Xi_{33} & \Xi_{23}+\Xi_{32} \\ -i\Xi_{12}+i\Xi_{21} & \Xi_{13}+\Xi_{31} & \Xi_{23}+\Xi_{32} & -\Xi_{11}-\Xi_{22}+\Xi_{33} \end{pmatrix}.
\end{aligned}\tag{65}
$$

Since $\Xi_{ij}$ is symmetric for $i, j$, it becomes

$$
= \begin{pmatrix} -\Xi_{11}-\Xi_{22}-\Xi_{33} & 0 & 0 & 0 \\ 0 & \Xi_{11}-\Xi_{22}-\Xi_{33} & 2\Xi_{12} & 2\Xi_{13} \\ 0 & 2\Xi_{12} & -\Xi_{11}+\Xi_{22}-\Xi_{33} & 2\Xi_{23} \\ 0 & 2\Xi_{13} & 2\Xi_{23} & -\Xi_{11}-\Xi_{22}+\Xi_{33} \end{pmatrix}.\tag{66}
$$

For Wely conformal spinor $\Psi_{ABCD} = \frac{1}{3}(X_{ABCD} + X_{ACDB} + X_{ADBC})$,

$$
\begin{aligned}
\Psi_{ABCD}&\bar{s}^{AC}_\mu\bar{s}^{BD}_\nu = \frac{1}{12}\Xi_{ij}(s^i_{AB}s^j_{CD} + s^i_{AC}s^j_{DB} + s^i_{AD}s^j_{BC})\bar{s}^{AC}_\mu\bar{s}^{BD}_\nu \\
&= \frac{1}{3}\begin{pmatrix} 0 & -i\Xi_{23}+i\Xi_{32} & i\Xi_{13}-i\Xi_{31} & -i\Xi_{12}+i\Xi_{21} \\ 0 & 2\Xi_{11}-\Xi_{22}-\Xi_{33} & 2\Xi_{12}+\Xi_{21} & 2\Xi_{13}+\Xi_{31} \\ 0 & \Xi_{12}+2\Xi_{21} & -\Xi_{11}+2\Xi_{22}-\Xi_{33} & 2\Xi_{23}+\Xi_{32} \\ 0 & \Xi_{13}+2\Xi_{31} & \Xi_{23}+2\Xi_{32} & -\Xi_{11}-\Xi_{22}+2\Xi_{33} \end{pmatrix}.
\end{aligned}\tag{67}
$$

Considering the symmetricity of $\Xi$, it becomes

$$
= \frac{1}{3}\begin{pmatrix} 0 & 0 & 0 & 0 \\ 0 & 2\Xi_{11}-\Xi_{22}-\Xi_{33} & 3\Xi_{12} & 3\Xi_{13} \\ 0 & 3\Xi_{12} & -\Xi_{11}+2\Xi_{22}-\Xi_{33} & 3\Xi_{23} \\ 0 & 3\Xi_{13} & 3\Xi_{23} & -\Xi_{11}-\Xi_{22}+2\Xi_{33} \end{pmatrix}.\tag{68}
$$

The components of $X_{ABCD}, \Psi_{ABCD}$ are expressed as symmetric tensors. As we can see on Equation (67) and Matrix (68), $\Xi_{ij}$ includes all information of $\Psi_{ABCD}$. Because of Wely tensor $C_{abcd} = \Psi_{ABCD}\varepsilon_{A'B'}\varepsilon_{C'D'} + \bar{\Psi}_{A'B'C'D'}\varepsilon_{AB}\varepsilon_{CD}$, we may conclude that all informations of Weyl tensor are comprehended in $\Xi_{ij}$.

The form of Matrix (68) is similar to the tidal tensor $\mathbb{T}_{ij}$ with a potential $U = -U_0/r = -U_0/\sqrt{x^2+y^2+z^2}$:

$$\mathbb{T}_{ij} = \frac{U_0}{r^5}\begin{pmatrix} 2x^2-y^2-z^2 & 3xy & 3xz \\ 3xy & -x^2+2y^2-z^2 & 3yz \\ 3xz & 3yz & -x^2-y^2+2z^2 \end{pmatrix},\tag{69}$$

where $\mathbb{T}_{ij} = J_{ij} - J^a_a\delta_{ij}$ and $J_{ij} = \delta^2 U/\delta x^i\delta x^j$ [27,28]. The similarity may come from the link between tidal forces and Weyl tensor. The tidal force in general relativity is described by the Riemann curvature tensor. The Riemman curvature tensor $R_{abcd}$ can be decomposed to $R_{abcd} = S_{abcd} + C_{abcd}$, where $C_{abcd}$

is a traceless part which is a Weyl tensor and $S_{abcd}$ is a remaining part which consists of Ricci tensor $R_{ab} = R^c{}_{acb}$ and $R = R^a_a$ [6]. In the Schwartzchild metric, since $R = R_{ab} = S_{abcd} = 0$ but $C_{abcd} \neq 0$, the tidal forces are described by Weyl tensor. This shows that $C_{abcd}$, $\Psi_{ABCD}$ and $\Xi_{ij}$ are all related to the tidal effects.

The components of $\Psi$ and $\Xi$ can be represented with Weyl tensors. In a flat coordinate, by using

$$R_{\mu\rho} = R_{\mu\nu\rho\sigma}g^{\nu\sigma} = R_{\mu0\rho0} - R_{\mu i\rho i} \tag{70}$$

and Equation (58), the components of Weyl tensor referred to as Equation (37) can be expressed as

$$C_{0\underline{p}0\underline{q}} = C_{\underline{j}\,\underline{p}\,\underline{j}\,\underline{q}} = \frac{1}{2}R_{0\underline{p}0\underline{q}} + \frac{1}{2}R_{\underline{p}j\underline{q}j} \quad (\text{for } p \neq q)\,, \tag{71}$$

$$C_{0\underline{p}0\underline{p}} = -C_{\underline{i}\,\underline{j}\,\underline{i}\,\underline{j}} = \frac{1}{2}R_{0\underline{p}0\underline{p}} - \frac{1}{2}R_{0k0k} + \frac{1}{2}R_{\underline{p}k\underline{p}k} - \frac{1}{2}R_{klkl}\,, \tag{72}$$

$$C_{\underline{p}0pq} = R_{\underline{p}0pq} - \frac{1}{2}R_{0i\underline{q}i}\,, \tag{73}$$

$$C_{\underline{i}0pq} = R_{\underline{i}0pq}\,. \tag{74}$$

Comparing Equations (71)—(74) with Equations (61) and (62), we find that

$$C_{0\underline{p}0\underline{q}} = C_{\underline{j}\,\underline{p}\,\underline{j}\,\underline{q}} = \frac{1}{2}E_{\underline{p}\underline{q}} \quad (\text{for } p \neq q)\,, \tag{75}$$

$$C_{0\underline{p}0\underline{p}} = -C_{\underline{i}\,\underline{j}\,\underline{i}\,\underline{j}} = \frac{1}{6}(3E_{\underline{p}\underline{p}} - E_{11} - E_{22} - E_{33})\,, \tag{76}$$

$$C_{\underline{p}0pq} = \epsilon_{\underline{i}\,\underline{p}\,q}\frac{Q_{\underline{i}\,\underline{p}}}{2}\,, \tag{77}$$

$$C_{\underline{i}0pq} = \epsilon_{\underline{i}\,\underline{p}\,q}\frac{Q_{ii}}{2} \quad (\text{for } i \neq p \text{ and } i \neq q)\,. \tag{78}$$

Therefore, Matrix (68) can be rewritten to

$$\Psi_{ij} = \Psi_{ABCD}\bar{s}_i^{AC}\bar{s}_j^{BD} = 2C_{0i0j} - i\epsilon^{ipq}C_{j0pq} + \frac{i}{3}\epsilon^{lpq}C_{l0pq}\,. \tag{79}$$

Since $\epsilon^{lpq}C_{l0pq} = Q_{11} + Q_{22} + Q_{33}$ is zero by Bianchi identity, it becomes

$$\Psi_{ij} = \Psi_{ABCD}\bar{s}_i^{AC}\bar{s}_j^{BD} = 2C_{0i0j} - i\epsilon^{ipq}C_{j0pq}\,. \tag{80}$$

Equation (63) can be reformulated to

$$\Xi_{ij} = 2C_{0i0j} - \frac{R}{6}\delta_{ij} - i\epsilon^{ipq}C_{j0pq}\,, \tag{81}$$

where $R = -2E_{ii} = -2\Xi_{ii} = -2(E_{11} + E_{22} + E_{33})$. Therefore, we can finally find the relation

$$\Xi_{ij} = \Psi_{ij} - \frac{R}{6}\delta_{ij}. \tag{82}$$

Here we can see the equivalence and the direct correspondences among $\Psi_{ABCD}$, $\Xi_{ij}$ and Weyl tensor.

## 5. Definition of Sedon and Relations among Spinors, Sedenion and Sedon

In this section, we investigate the basis of sedenion and we define a new algebra which is a similar structure to sedenion. Sedenion is 16 dimensional noncommutative and nonassociative algebra, which can be obtained from Cayley–Dickson construction [29,30]. The multiplication table of sedenion basis is shown in Table 2. The elements of sedenion basis can be represented in the form $e_i = q^\mu \otimes q'^{\mu'} = q^{\mu\mu'}$

with $i = \mu + 4\nu$, where $q^{\mu} = (1, \mathbf{i}, \mathbf{j}, \mathbf{k})$, $q'^{\mu'} = (1, \mathbf{i}', \mathbf{j}', \mathbf{k}')$. The multiplication rule can be written by $e_i * e_j = q^{\mu\mu'} * q^{\nu\nu'} = s_{\mu\mu'\nu\nu'} \, q^{\mu\mu'} q^{\nu\nu'}$, where $s_{\mu\mu'\nu\nu'}$ is +1 or −1, which is determined by $\mu, \mu', \nu, \nu'$ [7].

**Table 2.** The multiplication table of sedenion. For convenience, '$e_N$'s are represented as '$eN$' ; e.g., $e_3 \to$ e3.

| | e0 | e1 | e2 | e3 | e4 | e5 | e6 | e7 | e8 | e9 | e10 | e11 | e12 | e13 | e14 | e15 |
|---|---|---|---|---|---|---|---|---|---|---|---|---|---|---|---|---|
| **e0** | e0 | e1 | e2 | e3 | e4 | e5 | e6 | e7 | e8 | e9 | e10 | e11 | e12 | e13 | e14 | e15 |
| **e1** | e1 | −e0 | e3 | −e2 | e5 | −e4 | −e7 | e6 | e9 | −e8 | −e11 | e10 | −e13 | e12 | e15 | −e14 |
| **e2** | e2 | −e3 | −e0 | e1 | e6 | e7 | −e4 | −e5 | e10 | e11 | −e8 | −e9 | −e14 | −e15 | e12 | e13 |
| **e3** | e3 | e2 | −e1 | −e0 | e7 | −e6 | e5 | −e4 | e11 | −e10 | e9 | −e8 | −e15 | e14 | −e13 | e12 |
| **e4** | e4 | −e5 | −e6 | −e7 | −e0 | e1 | e2 | e3 | e12 | e13 | e14 | e15 | −e8 | −e9 | −e10 | −e11 |
| **e5** | e5 | e4 | −e7 | e6 | −e1 | −e0 | −e3 | e2 | e13 | −e12 | e15 | −e14 | e9 | −e8 | e11 | −e10 |
| **e6** | e6 | e7 | e4 | −e5 | −e2 | e3 | −e0 | −e1 | e14 | −e15 | −e12 | e13 | e10 | −e11 | −e8 | e9 |
| **e7** | e7 | −e6 | e5 | e4 | −e3 | −e2 | e1 | −e0 | e15 | e14 | −e13 | −e12 | e11 | e10 | −e9 | −e8 |
| **e8** | e8 | −e9 | −e10 | −e11 | −e12 | −e13 | −e14 | −e15 | −e0 | e1 | e2 | e3 | e4 | e5 | e6 | e7 |
| **e9** | e9 | e8 | −e11 | e10 | −e13 | e12 | e15 | −e14 | −e1 | −e0 | −e3 | e2 | −e5 | e4 | e7 | −e6 |
| **e10** | e10 | e11 | e8 | −e9 | −e14 | −e15 | e12 | e13 | −e2 | e3 | −e0 | −e1 | −e6 | −e7 | e4 | e5 |
| **e11** | e11 | −e10 | e9 | e8 | −e15 | e14 | −e13 | e12 | −e3 | −e2 | e1 | −e0 | −e7 | e6 | −e5 | e4 |
| **e12** | e12 | e13 | e14 | e15 | e8 | −e9 | −e10 | −e11 | −e4 | e5 | e6 | e7 | −e0 | −e1 | −e2 | −e3 |
| **e13** | e13 | −e12 | e15 | −e14 | e9 | e8 | e11 | −e10 | −e5 | −e4 | e7 | −e6 | e1 | −e0 | e3 | −e2 |
| **e14** | e14 | −e15 | −e12 | e13 | e10 | −e11 | e8 | e9 | −e6 | −e7 | −e4 | e5 | e2 | −e3 | −e0 | e1 |
| **e15** | e15 | e14 | −e13 | −e12 | e11 | e10 | −e9 | e8 | −e7 | e6 | −e5 | −e4 | e3 | e2 | −e1 | −e0 |

Table 3 shows the multiplication table of an algebra which is similar to sedenion. It consists of 16 bases $e_i = q^{\mu} \otimes q'^{\mu'}$ with $i = \mu + 4\nu$ and the multiplication rule $e_i * e_j = (q^{\mu} \otimes q'^{\mu'}) * (q^{\mu} \otimes q'^{\mu'}) = (q^{\mu}q^{\mu} \otimes q'^{\mu'}q'^{\mu'})$. The table is almost the same as the multiplication table of sedenion basis, but just differs in signs. The signs of red colored elements in Table 3 differ from Table 2. We will call this algebra as 'sedon'.

Sedon can be written in the form

$$S = A_0 + |\vec{B}\} + \{\vec{C}| + \{\overleftrightarrow{D}\} = A_0 + B_i q_R^i + C_i q_L^i + D_{ij} u^{ij}, \tag{83}$$

where $q_R^i = 1 \otimes q^i$, $q_L^i = q^i \otimes 1$, $u^{ij} = q^i \otimes q^j$, $|\vec{B}\} = B_i q_R^i$, $\{\vec{C}| = C_i q_L^i$, and $\{\overleftrightarrow{D}\} = D_{ij} u^{ij}$. We can name $|\vec{B}\}$ as 'right svector', $\{\vec{C}|$ as 'left svector', and $\{\overleftrightarrow{D}\}$ as 'stensor'. The coefficient of sedon can be represented as in Table 4. For example, $D_{13}$ is a coefficient of $i \otimes k$ term.

Now we will see the relation between Ricci spinors and the sedon. Since

$$\sigma_A^{i\,C} \varepsilon_{CB} = s_{AB}^i, \qquad \varepsilon_{A'C'} \sigma_{B'}^{C'} = -\bar{s}_{A'B'}^i, \tag{84}$$

therefore

$$\sigma_A^{i\,B} = -s_{AC}^i \varepsilon^{CB}, \qquad \sigma_{B'}^{i\,C'} = \varepsilon^{C'A'} \bar{s}_{A'B'}^i. \tag{85}$$

Equation (50) can be reformulated as

$$\Phi_A{}^{QP'}{}_{D'} = \varepsilon^{P'C'} \Phi_{ABC'D'} \varepsilon^{BQ} = \frac{1}{4}\Theta_{ij} \varepsilon^{P'C'} s_{AB}^i \bar{s}_{C'D'}^j \varepsilon^{BQ} = -\frac{1}{4}\Theta_{ij} \sigma_A^{i\,Q} \sigma_{D'}^{j\,P'}. \tag{86}$$

Since $q^i = (\mathbf{i}, \mathbf{j}, \mathbf{k})$ is isomorphic to $-i\sigma^i = (-\sigma^1 i, -\sigma^2 i, -\sigma^3 i)$, we can set $q^i = -i\sigma^i$. Equation (86) can be written as

$$\Phi_A{}^{QP'}{}_{D'} = \frac{1}{4}\Theta_{ij} q_A^{i\,Q} q_{D'}^{j\,P'}, \tag{87}$$

which can be regarded as a sedon. In a similar way, $X_{ABCD}$ can be written as

$$X_A{}^B{}_C{}^D = \frac{1}{4}\Xi_{kl} q_A^{k\,B} q_C^{l\,D}. \tag{88}$$

From Equation (87), a Ricci spinor can be interpreted as a combination of a right-handed and a left-handed rotational operations, since the basis has the form 'left-handed quaternion $\otimes$ right- handed quaternion'. Following the rotational interpretation of Cayley–Dickson algebra [7], it can be interpreted as the twofold rotation $\otimes$ twofold rotation.

For two quaternions $Ą = A_i q^i = a_1\mathbf{i} + a_2\mathbf{j} + a_3\mathbf{k}$ and $Ɓ = B_j q^j = b_1\mathbf{i} + b_2\mathbf{j} + b_3\mathbf{k}$, which can be represented in the $2 \times 2$ matrix representation with spinor indices ($A_i q^i{}_C{}^D$ and $B_j q^j{}_C{}^D$), the multiplication of them can be written as

$$Ą Ɓ = A_i q^i{}_C{}^D\, B_j q^j{}_D{}^E = -A_i B_i \delta_C{}^E + \epsilon_{ijk} A_i B_j q^k{}_C{}^E. \tag{89}$$

We can use this to express multiplications of spinors. One of the example is

$$
\begin{aligned}
\Phi_A{}^{BC'}{}_{D'} \Phi_B{}^{ED'}{}_{F'} \quad &= \theta_{ij} q^i{}_A{}^B q^j{}^{C'}{}_{D'}\, \theta_{rs} q^r{}_B{}^E q^s{}^{D'}{}_{F'} \\
&= (-\theta_{lj}\theta_{ls}\delta_A{}^E + \epsilon_{pqu}\theta_{pj}\theta_{qs} q^u{}_A{}^E) q^j{}^{C'}{}_{D'} q^s{}^{D'}{}_{F'} \\
&= \theta_{lk}\theta_{lk}\ \delta_A{}^E \delta^{C'}{}_{F'} - \epsilon_{mnv}\theta_{lm}\theta_{ln}\ q^v{}^{C'}{}_{F'}\delta_A{}^E \\
&\quad - \epsilon_{pqu}\theta_{pl}\theta_{ql}\ q^u{}_A{}^E \delta^{C'}{}_{F'} + \epsilon_{mnv}\epsilon_{pqu}\theta_{pm}\theta_{qn}\ q^u{}_A{}^E q^v{}^{C'}{}_{F'},
\end{aligned}
\tag{90}
$$

where $\theta_{ij} = \frac{1}{4}\Theta_{ij}$. The result is also a sedon form. Above example shows not only multiplications of $\Phi_A{}^{BC'}{}_{D'}$, but also the general multiplication of stensor. Here is an another example: An antisymmetric differential operator $\nabla_{[a}\nabla_{b]}$ can be divided into two parts

$$\Delta_{ab} = 2\nabla_{[a}\nabla_{b]} = \epsilon_{A'B'}\square_{AB} + \epsilon_{AB}\square_{A'B'}, \tag{91}$$

where $\square_{AB} = \frac{1}{2}\Delta_{AA'B}{}^{A'}$ and $\square_{A'B'} = \frac{1}{2}\Delta_{AA'}{}^A{}_{B'}$. As we can see in Equations (9) and (11), each term can be considered as a quaternion.

$$\square_A{}^B = \frac{1}{4}\Delta_{\mu\nu}\sigma^\mu_{AA'}\bar{\sigma}^{\nu\,A'B} = \frac{1}{2}\left(i\,\Delta_{k0} + \frac{1}{2}\epsilon^{ij}{}_k\Delta_{ij}\right) q^k{}_A{}^B \tag{92}$$

$$\bar{\square}^{A'}{}_{B'} = -\frac{1}{4}\Delta_{\mu\nu}\,\bar{\sigma}^{\mu A'C}\sigma^\nu_{CB'} = \frac{1}{2}\left(i\,\Delta_{k0} - \frac{1}{2}\epsilon^{ij}{}_k\Delta_{ij}\right) q^k{}^{A'}{}_{B'}. \tag{93}$$

Then, $\square_A{}^B \Phi_B{}^{CD'}{}_{E'}$ can be considered as a multiplication of a quaternion and a sedon.

$$
\begin{aligned}
\square_A{}^B \Phi_B{}^{CD'}{}_{E'} \quad &= ♭_k\, q^k{}_A{}^B \theta_{ij} q^i{}_B{}^C q^j{}^{D'}{}_{E'} \\
&= -♭_p\, \theta_{pj}\delta_A{}^C q^j{}^{D'}{}_{E'} + \epsilon_{kip}♭_k\, \theta_{ij} q^p{}_A{}^C q^j{}^P{}_T \\
&= -i\,\Delta_{p0}\theta_{pj}\ \delta_A{}^C q^j{}^{D'}{}_{E'} - \frac{1}{2}\epsilon_{rsp}\Delta_{rs}\theta_{pj}\ \delta_A{}^C q^j{}^{D'}{}_{E'} \\
&\quad + i\,\epsilon_{kip}\Delta_{k0}\theta_{ij}\ q^p{}_A{}^C q^j{}^{D'}{}_{E'} + \frac{1}{2}\epsilon_{lqk}\Delta_{lq}\epsilon_{kip}\theta_{ij}\ q^p{}_A{}^C q^j{}^{D'}{}_{E'},
\end{aligned}
\tag{94}
$$

where $♭_k \equiv i\Delta_{k0} + \frac{1}{2}\epsilon_{ijk}\Delta_{ij}$. $\epsilon_{lqk}\Delta_{lq}\epsilon_{kip}\theta_{ij}$ in the last term can be changed as $\Delta_{qp}\theta_{qj} - \Delta_{pq}\theta_{qj} = 2\Delta_{qp}\theta_{qj}$. The result in Equation (94) is in a sedon form. Using those expressions, we can represent the quantities with spinor indices as sedon forms whose elements are components of tensors.

**Table 3.** The multiplication table of sedon.

|     | e0  | e1  | e2   | e3   | e4   | e5   | e6   | e7   | e8   | e9   | e10  | e11  | e12  | e13  | e14  | e15  |
|-----|-----|-----|------|------|------|------|------|------|------|------|------|------|------|------|------|------|
| e0  | e0  | e1  | e2   | e3   | e4   | e5   | e6   | e7   | e8   | e9   | e10  | e11  | e12  | e13  | e14  | e15  |
| e1  | e1  | −e0 | e3   | −e2  | e5   | −e4  | e7   | −e6  | e9   | −e8  | e11  | −e10 | e13  | −e12 | e15  | −e14 |
| e2  | e2  | −e3 | −e0  | e1   | e6   | −e7  | −e4  | e5   | e10  | −e11 | −e8  | e9   | e14  | −e15 | −e12 | e13  |
| e3  | e3  | e2  | −e1  | −e0  | e7   | e6   | −e5  | −e4  | e11  | e10  | −e9  | −e8  | e15  | e14  | −e13 | −e12 |
| e4  | e4  | e5  | e6   | e7   | −e0  | −e1  | −e2  | −e3  | e12  | e13  | e14  | e15  | −e8  | −e9  | −e10 | −e11 |
| e5  | e5  | −e4 | e7   | −e6  | −e1  | e0   | −e3  | e2   | e13  | −e12 | e15  | −e14 | −e9  | e8   | −e11 | e10  |
| e6  | e6  | −e7 | −e4  | e5   | −e2  | e3   | e0   | −e1  | e14  | −e15 | −e12 | e13  | −e10 | e11  | e8   | −e9  |
| e7  | e7  | e6  | −e5  | −e4  | −e3  | −e2  | e1   | e0   | e15  | e14  | −e13 | −e12 | −e11 | −e10 | e9   | e8   |
| e8  | e8  | e9  | e10  | e11  | −e12 | −e13 | −e14 | −e15 | −e0  | −e1  | −e2  | −e3  | e4   | e5   | e6   | e7   |
| e9  | e9  | −e8 | e11  | −e10 | −e13 | e12  | −e15 | e14  | −e1  | e0   | −e3  | e2   | e5   | −e4  | e7   | −e6  |
| e10 | e10 | −e11| −e8  | e9   | −e14 | e15  | e12  | −e13 | −e2  | e3   | e0   | −e1  | e6   | −e7  | −e4  | e5   |
| e11 | e11 | e10 | −e9  | −e8  | −e15 | −e14 | e13  | e12  | −e3  | −e2  | e1   | e0   | e7   | e6   | −e5  | −e4  |
| e12 | e12 | e13 | e14  | e15  | e8   | e9   | e10  | e11  | −e4  | −e5  | −e6  | −e7  | −e0  | −e1  | −e2  | −e3  |
| e13 | e13 | −e12| e15  | −e14 | e9   | −e8  | e11  | −e10 | −e5  | e4   | −e7  | e6   | −e1  | e0   | −e3  | e2   |
| e14 | e14 | −e15| −e12 | e13  | e10  | −e11 | −e8  | e9   | −e6  | e7   | e4   | −e5  | −e2  | e3   | e0   | −e1  |
| e15 | e15 | e14 | −e13 | −e12 | e11  | e10  | −e9  | −e8  | −e7  | −e6  | e5   | e4   | −e3  | −e2  | e1   | e0   |

**Table 4.** The representation of coefficients of sedon.

|                     | $\sim \otimes \mathbf{1}$ | $\sim \otimes \mathbf{i}$ | $\sim \otimes \mathbf{j}$ | $\sim \otimes \mathbf{k}$ |
|---------------------|:-------------------------:|:-------------------------:|:-------------------------:|:-------------------------:|
| $\mathbf{1} \otimes \sim$ | $A_0$    | $B_1$     | $B_1$     | $B_1$     |
| $\mathbf{i} \otimes \sim$ | $C_1$    | $D_{11}$  | $D_{12}$  | $D_{13}$  |
| $\mathbf{j} \otimes \sim$ | $C_2$    | $D_{21}$  | $D_{22}$  | $D_{23}$  |
| $\mathbf{k} \otimes \sim$ | $C_3$    | $D_{31}$  | $D_{32}$  | $D_{33}$  |

## 6. A Few Examples of Curvature Spinors in a Locally Flat Coordinate

### 6.1. Weyl Conformal Spinor for the Schwarzschild Metric: An Example of Section IV

It is known that the Schwarzschild metric can be represented in Fermi normal coordinate [31]. In Schwarzschild coordinate $x^{\mu'} = (T, R, \Theta, \Phi)$, the metric is displayed in the form

$$ds^2 = g_{\mu'\nu'} dy^{\mu'} dy^{\nu'} = -f dT^2 + f^{-1} dR^2 + R^2 d\Theta^2 + R^2 \sin^2\Theta\, d\Phi^2 , \tag{95}$$

where $f = 1 - 2GM/R$. The basis of a constructed Fermi coordinate $x^\mu = (t, x, y, z)$ is

$$
\begin{aligned}
\mathbf{e}_0 \quad &= \partial/\partial t|_G = T'\, \partial/\partial T + R'\, \partial/\partial R, \\
\mathbf{e}_1 \quad &= \partial/\partial x|_G = f^{-1} R'\, \partial/\partial T + f T'\, \partial/\partial R, \\
\mathbf{e}_2 \quad &= \partial/\partial y|_G = 1/R\, \partial/\partial\Theta, \\
\mathbf{e}_3 \quad &= \partial/\partial z|_G = 1/R\, \sin\Theta\, \partial/\partial\Phi,
\end{aligned}
\tag{96}
$$

where the primes indicate derivatives with respect to proper time $t$. The non-zero components of the Riemann curvature tensor $R_{\mu'\nu'\rho'\sigma'}$ in Schwarzschild coordinate are

$$
\begin{aligned}
R_{1'0'1'0'} \quad &= 2GM/R^2, \\
R_{3'0'3'0'} \quad &= -(fGM/R)\sin^2\Theta, \\
R_{1'2'1'2'} \quad &= GM/(fR), \\
R_{2'0'2'0'} \quad &= -fGM/R, \\
R_{2'3'2'3'} \quad &= -2GMR\sin^2\Theta, \\
R_{1'3'1'3'} \quad &= (GM/Rf)\sin^2\Theta.
\end{aligned}
\tag{97}
$$

Then the Riemman curvature tenor $R_{\mu\nu\rho\sigma}$ in the Fermi coordinate is

$$
\begin{aligned}
R_{10\,10} \quad &= 2GM/R^3, \\
R_{20\,20} \quad &= R_{30\,30} = -GM/R^3,
\end{aligned}
$$

$$
\begin{aligned}
R_{12\,12} &= R_{13\,13} = GM/R^3, \\
R_{23\,23} &= -2GM/R^3.
\end{aligned}
\tag{98}
$$

From Equations (47), (48), (61) and (62), we can observe that $P_{ij} = S_{ij} = Q_{ij} = 0$, but $E_{11} = 4GM/R^3$ and $E_{22} = E_{33} = -2GM/R^3$. Classically, the tidal acceleration of black hole along the radial line is $-2GM/r^3\delta X$, and the acceleration perpendicular to the radial line is $GM/r^3\delta X$, where $\delta X$ is the separation distance of two test particles. In this example, the link between $\Xi_{ij}$ and tidal accelerations has been shown.

*6.2. The Spinor Form of the Einstein–Maxwell Equation: An Example of Section V*

Einstein–Maxwell Equations, which is Einstein Equations in presence of electromagnetic fields, is known [32] as

$$
R_{\mu\nu} - \frac{1}{2}g_{\mu\nu}R = 8\pi G(F_{\mu\sigma}F_\nu^\sigma - g_{\mu\nu}\frac{1}{4}F_{\rho\sigma}F^{\rho\sigma}),
\tag{99}
$$

where $F_{\mu\sigma}F_\nu^\sigma - g_{\mu\nu}\frac{1}{4}F_{\rho\sigma}F^{\rho\sigma}$ is the electromagnetic stress-energy tensor. The spinor form of the Einstein–Maxwell equation [1] is

$$
\Phi_{ABA'B'} = 8\pi G\varphi_{AB}\bar\varphi_{A'B'},
\tag{100}
$$

where $\varphi_{AB}, \bar\varphi_{A'B'}$ are decomposed spinors of electromagnetic tensor $F_{\mu\nu}$, as following Equations (18) and (19). This can be deformed to

$$
\Phi_A{}^{BA'}{}_{B'} = 8\pi G\varphi_A{}^B\bar\varphi{}^{A'}{}_{B'}.
\tag{101}
$$

From Equation (86),

$$
\begin{aligned}
&-\frac{1}{4}\Theta_{kl}\sigma_A{}^{k\,B}\sigma^{l\,A'}{}_{B'} \\
&= 8\pi G[\frac{1}{2}(F_{k0} - \frac{1}{2}i\,\epsilon^{ij}{}_k F_{ij})\sigma_A{}^{k\,B}] \times [\frac{1}{2}(F_{l0} + \frac{1}{2}i\,\epsilon^{pq}{}_l F_{pq})\sigma^{l\,A'}{}_{B'}] \\
&= 2\pi G[(F_{k0}F_{l0} + \frac{1}{4}\epsilon^{ij}{}_k\epsilon^{pq}{}_l F_{ij}F_{pq}) + \frac{i}{2}(F_{k0}\epsilon^{pq}{}_l F_{pq} - F_{l0}\epsilon^{ij}{}_k F_{ij})]\sigma_A{}^{k\,B}\sigma^{l\,A'}{}_{B'}.
\end{aligned}
\tag{102}
$$

Comparing the first line with the third line in Equation (102), we get

$$
S_{kl} = -8\pi G(F_{k0}F_{l0} + \frac{1}{4}\epsilon^{ij}{}_k\epsilon^{pq}{}_l F_{ij}F_{pq}),
\tag{103}
$$

$$
P_{kl} = -4\pi G(F_{k0}\epsilon^{pq}{}_l F_{pq} - F_{l0}\epsilon^{ij}{}_k F_{ij}),
\tag{104}
$$

and, from Equation (104) we get

$$
\epsilon^{mkl}P_{kl} = -16\pi G F_{k0}F^{mk}.
\tag{105}
$$

For $F^{\mu\nu}$ such that

$$
F^{\mu\nu} = \begin{pmatrix} 0 & -E_1 & -E_2 & -E_3 \\ E_1 & 0 & -B_3 & B_2 \\ E_2 & B_3 & 0 & -B_1 \\ E_3 & -B_2 & B_1 & 0 \end{pmatrix},
\tag{106}
$$

we have $F_{k0}F^{mk} = (\vec{E} \times \vec{B})^m$ and $(F_{k0}F_{l0} + \frac{1}{4}\epsilon^{ij}{}_k\epsilon^{pq}{}_l F_{ij}F_{pq}) = E_k E_l + B_k B_l$. From Equations (51) and (104) we get

$$-\frac{1}{2}\epsilon^{mkl}P_{kl} \quad = R_{0i}{}^{mi} = 8\pi G(\vec{E} \times \vec{B})^k \,. \tag{107}$$

This is a momentum of electromagnetic tensor and it shows that $P_{ij}/(8\pi G)$ is related to momentum. From Equations (49) and (103),

$$S_{kl} \quad = R_{0i\,0j} + \frac{1}{4}\epsilon_{pqi}\epsilon_{rsj}R_{pq\,rs} = 8\pi G(-E_k E_l - B_k B_l), \tag{108}$$

$$S_{ll} \quad = 8\pi G(|\vec{E}|^2 + |\vec{B}|^2). \tag{109}$$

Those are the shear stress and the energy of electromagnetic field. It shows that $S_{ij}/(8\pi G)$ is related to stress-energy.

### 6.3. The Quaternion form of Differential Bianchi Identity: Another Example of Section V

The spinor form of Bianchi identity referred to as Equation (27) is known [1] as

$$\nabla^A_{B'}X_{ABCD} = \nabla^{A'}_B\Phi_{CDA'B'}, \tag{110}$$

which can be deformed to

$$\nabla^{B'A}X_A{}^B{}_C{}^D = \nabla^{BA'}\Phi_C{}^D{}_{A'}{}^{B'}. \tag{111}$$

In flat coordinate, $\nabla^{A'A}$ equals to

$$\partial^{A'A} = g^{\mu\,A'A}\partial_\mu = \frac{1}{\sqrt{2}}\sigma^{\mu A'A}\partial_\mu = \frac{1}{\sqrt{2}}\sigma^{\tilde{\mu}A'A}\partial_{\tilde{\mu}}$$

$$= \frac{1}{\sqrt{2}}\bar{q}^{\mu\,A'A}\tilde{\partial}_\mu = \frac{1}{\sqrt{2}}q^{\mu\,A'A}\partial'_\mu \,, \tag{112}$$

where $\tilde{\mu}$ is tilde-spacetime indices which is defined as $O^{\tilde{\mu}} = (O^0, iO^1, iO^2, iO^3)$, $O_{\tilde{\mu}} = (O_0, -iO_1, -iO_2, -iO_3)$ for any $O^\mu = (O^0, O^1, O^2, O^3)$, $O_\mu = (O_0, O_1, O_2, O_3)$ [7]; $\bar{q}^\mu$ is $\bar{q}^\mu = \sigma^{\tilde{\mu}A'A} = (\sigma^0, i\sigma^1, i\sigma^2, i\sigma^3)$ which is isomorphic to $(1, -\mathbf{i}, -\mathbf{j}, -\mathbf{k})$, $\tilde{\partial}_\mu = \partial_{\tilde{\mu}} = (\partial_0, -i\partial_1, -i\partial_2, -i\partial_3)$, and $\partial'_\mu = (\partial_0, i\partial_1, i\partial_2, i\partial_3)$. We used the property $A_\mu B^\mu = A_{\tilde{\mu}}B^{\tilde{\mu}}$ [7]. $\partial^{A'A}$ can be expanded to

$$\partial^{A'A} \quad = \partial_0\delta^{A'A} + \partial'_k q^{k\,A'A} \tag{113}$$

and, considering matrix representation, $\partial^{AA'}$ can be represented as

$$\partial^{AA'} \quad = \partial_0\delta^{AA'} + \partial'_k q^{\bar{k}\,AA'} = \partial_0\delta^{AA'} + \partial'_{\bar{k}}q^{k\,AA'} \,, \tag{114}$$

where the bar index $A^{\bar{k}}$ means the opposite-handed quantity of $A^k$, which is $A^{\bar{1}} = A^1, A^{\bar{2}} = -A^2, A^{\bar{3}} = A^3$ ; when $k = 2$, $\bar{k}$ index change sings of $A^k$. It has following properties,

$$A^{\bar{k}}B_k = A^k B_{\bar{k}}, \quad A^{\bar{k}}B_{\bar{k}} = A^k B_k, \quad \varepsilon_{pqr}A^{\bar{q}}B^{\bar{r}} = -\varepsilon_{\bar{p}qr}A^q B^r, \quad \varepsilon_{\bar{p}\bar{q}\bar{r}} = -\varepsilon_{pqr}. \tag{115}$$

Then Equation (111) can be written as

$$(\partial_0\delta^{B'A} + \partial'_k q^{k\,B'A})\,\Xi_{ir}q^i{}_A{}^B q^r{}_C{}^D = (\partial_0\delta^{BA'} + \partial'_{\bar{k}}q^{k\,BA'})\,\Theta_{r\bar{s}}q^r{}_C{}^D q^s{}_{A'}{}^{B'}, \tag{116}$$

since $q^{\bar{s}\;B'}_{\;A'} = \varepsilon^{B'D'}\varepsilon_{C'A'}q^{s\;C'}_{\;D'}$. Using Equation (89),

$$\partial_0 \Xi_{ir}q^{i\;B'B} + \partial'_k\Xi_{kr}\delta^{B'B} + \varepsilon_{pki}\partial'_k\Xi_{ir}q^{p\;B'B}$$
$$= \partial_0\Theta_{rs}q^{\bar{s}\;BB'} + \partial'_k\Theta_{rk}\delta^{BB'} + \varepsilon_{pks}\partial'_{\bar{k}}\Theta_{r\bar{s}}q^{p\;B'B}\;, \tag{117}$$

which can be rearranged as

$$\partial'_k(\Xi_{kr} - \Theta_{rk})\delta^{B'B} + [\partial_0(\Xi_{sr} - \Theta_{rs}) + \varepsilon_{ski}\partial'_k(\Xi_{ir} + \Theta_{ri})]q^{s\;B'B} = 0\;. \tag{118}$$

This is the quaternion form of Bianchi identities. Using $\Theta_{ij} = \bar{\Theta}_{ji}$ and denoting $\Xi_{ij}$ as $\Xi_j$, Equation (118) can be written as

$$i\,\nabla\cdot(\Xi - \bar{\Theta})_r - i\,\sigma\cdot(\Xi - \bar{\Theta})_r + \sigma\cdot(\nabla\times\Xi + \nabla\times\bar{\Theta})_r = 0\;, \tag{119}$$

where $\sigma$ are Pauli matrices.

*6.4. Spatial-Handedness of Graviton and Spin of Matters in Gravitational Phenomena*

Considering $q^k_{\;A}{}^B$,

$$q^{k\;A}_{\;B} = -\varepsilon^{AC}i\sigma^k_{\;C}{}^D\varepsilon_{DB} = -i(\sigma_1, -\sigma_2, \sigma_3)^A_{\;B}, \tag{120}$$

which can be understood as a spatially left-handed quaternion basis $q^{\bar{k}\;A}_{\;B}$, interchanging of up-down positions for spinor indices of quaternion basis indicates the interchanging of spatial-handedness of spatial index. As examples, Equations (87) and (88) can also be written as

$$\Phi^A_{\;B}{}^{C'}_{\;D'} = \frac{1}{4}\Theta_{\bar{i}j}q^{i\;A}_{\;B}q^{j\;C'}_{\;D'}\;, \tag{121}$$

$$X^A_{\;B}{}^C_{\;D} = \frac{1}{4}\Xi_{\bar{k}\bar{l}}\,q^{k\;A}_{\;B}q^{j\;C}_{\;D}\;. \tag{122}$$

In modified gravity or quantum gravity [33,34], the action could include higher order terms like $R^2, R_{ab}R^{ab}, R_{abcd}R^{abcd}$. Therefore, we can also include terms like $X^2, \Phi^2, \Phi X\Phi$... into the action of gravity. In such a case, following the representation like Equations (87) and (88) to couple those spinors together, the changes of spartial-handedness could occur, which means that spartial-right-handed particles couple to the spartial-left-handed and vice versa. For an example,

$$X^2 = X^B_{\;A}{}^D_{\;C}X^A_{\;B}{}^C_{\;D} = (\frac{1}{4}\Xi_{ij}\,q^{i\;B}_{\;A}q^{j\;D}_{\;C})(\frac{1}{4}\Xi_{kl}q^{\bar{k}\;A}_{\;B}q^{\bar{l}\;C}_{\;D}) \tag{123}$$

$$= \Xi_{ij}\Xi_{\bar{i}\bar{j}}. \tag{124}$$

This can affect the drawing of Feynman diagrams of graviton. When matters couple with spinors, it can be thought that spins of matter fields contribute to make a gravitational effects on matter, since $q^i_{\;A}{}^B\psi^A$ gives the spin of $\psi^A$.

## 7. Conclusions

We established a new method to express curvature spinors, which allows us to grasp components of the spinors easily in a locally inertial frame. During such a process, we technically utilized modified sigma matrices as a basis, which are sigma matrices multiplied by $\varepsilon$, and calculated the product of sigma matrices with mixed spinor indices. Using those modified sigma matrices as a basis can be regarded as

the rotation of the basis of four sigma matrices $(\sigma^0, \sigma^1, \sigma^2, \sigma^3)$ to $(s^0, s^1, s^2, s^3)$ defined in Equation (14), similar to a rotation of quaternion basis as shown in our previous work [7]. By comparing the Ricci spinor with the spinor form of Einstein equation, we could appreciate the roles of each component of the Riemann curvature tensor. The newly defined (3,3) tensors related to curvature tensors were introduced, and furthermore, from the representation of Weyl conformal spinor, we find that the components of Weyl tensor can be replaced by complex quantities $\Xi_{ij}$, which are defined in Equation (63). We represented the elements of sedenion basis as the direct product of elements of the quaternion bases themselves. Then we defined a new algebra 'sedon', which has the same basis representation except for a slightly modified multiplication rule from the multiplication rule of sedenion. The relations between sedon and the curvature spinors are derived for a general gravitational field, not just for a weak gravitational field. We calculated multiplications of spinors with a quaternion form, and observed that the results of the multiplications are also represented in a sedon form. The relations among quaternion, sedon and curvature spinors may imply that gravity could be the consequence of combination of right-handed and left-handed abstract rotational operations.

A few applications of the sedon representations were also introduced. One of the applications represented the Bianchi identity in the quaternion form, which might give the fluidic interpretation of the identity. We also suggested further possible application in modified gravity and quantum gravity. This may suggest that the spin-handedness, spatial-handedness, and spins of matter field would affect gravitational phenomena. The handedness structure of gravitational force has not yet been considered seriously. The sedon representation can be primarily a tool in which the handedness in gravity could be considered in detail. Finally we note that the research of this paper can be extended to general coordinates by considering vielbein formalism [35] and/or by establishing a connection to either self-dual or anti-self-dual variables [36,37].

**Author Contributions:** Conceptualization, I.K.H. and C.S.K.; investigation, I.K.H.; validation, I.K.H. and G.H.M.; writing—original draft preparation, I.K.H.; writing—review and editing, C.S.K. and G.H.M.; supervision, C.S.K.; funding acquisition, C.S.K.

**Funding:** This work was supported by the National Research Foundation of Korea (NRF) grant funded by the Korean government (MSIP) (NRF-2018R1A4A1025334).

**Acknowledgments:** This work was supported by the National Research Foundation of Korea (NRF) grant funded by the Korean government (MSIP) (NRF-2018R1A4A1025334).

**Conflicts of Interest:** The authors declare no conflict of interest.

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
