# Peer review of "Curvature Spinors in Locally Inertial Frame and the Relations with Sedenion"

_universe, doi:10.3390/universe6030040_

Round 1

Reviewer 1 Report

This paper deals with the spinor formalism for Lorentzian manifolds. In particular, alternative expressions for the spinorial representations of the curvature tensor using a three dimensional basis are provided and the relation between the curvature spinors and the sedonion is established. The paper is interesting and I think the material contained in it can be published. However, there are some points that should be clarified by the authors before publication.

* The paper is mostly written with respect to locally flat coordinates. Why is it necessary to resort to Fermi coordinates when working on a curved manifold if we can use instead a non-coordinate orthonormal basis? What can we say about your results if the coordinate frame is arbitrary?

* General relativity can be described using either self-dual or anti-self-dual variables (see for instance Plebanski, JMP 18, 2511 (1977); Capovilla et al CQG 8, 41 (1991)). It would be nice to establish a connection with those works.

* I do not understand why your approach suggest a reinterpretation of time. All you did was to split spacetime indices into time and spatial ones. No information has been lost. What is true is that the approach has lost its manifest spacetime covariance, but it is still there. Why should we reinterpret time?

* Regarding the notation, you sometimes use Greek letters and latin letters at the beginning of the alphabet to denote spacetime indices. I think a clear distiction of the the indices, if any, should be stated.

* In Eq. (96), what is the X appearing in e_1? I think it should be f of Eq. (95). Then, what is the X appearing in the paragraph under Eq. (98)?

* The manuscript is plagued with typos (mainly misspelled words). The authors should re-read it and correct them.

Author Response

Answers to Referee 1

This paper deals with the spinor formalism for Lorentzian manifolds. In particular, alternative expressions for the spinorial representations of the curvature tensor using a three dimensional basis are provided and the relation between the curvature spinors and the sedonion is established. The paper is interesting and I think the material contained in it can be published. However, there are some points that should be clarified by the authors before publication.

* The paper is mostly written with respect to locally flat coordinates. Why is it necessary to resort to Fermi coordinates when working on a curved manifold if we can use instead a non-coordinate orthonormal basis? What can we say about your results if the coordinate frame is arbitrary?

-->>We added comments about this part on page 4.

* General relativity can be described using either self-dual or anti-self-dual variables (see for instance Plebanski, JMP 18, 2511 (1977); Capovilla et al CQG 8, 41 (1991)). It would be nice to establish a connection with those works.

-->>We added a comment on the issue at the end of the conclusion​.

We added 3 new references, [35,36,37].

* I do not understand why your approach suggest a reinterpretation of time. All you did was to split spacetime indices into time and spatial ones. No information has been lost. What is true is that the approach has lost its manifest spacetime covariance, but it is still there. Why should we reinterpret time?

-->>The four directions time and space have changed into a bundle. We can choose arbitrary direction to be the 0-th coordinate (it does not have to be time direction) and the curvature spinors are represented in the base without 0-th components.  Technically, the interpretation of the time-space structure, which was considered to be a bundle of four directions, could be reconsidered.​ We added a comment in page 9.

* Regarding the notation, you sometimes use Greek letters and latin letters at the beginning of the alphabet to denote spacetime indices. I think a clear distiction of the the indices, if any, should be stated.

-->>We added  the statement of indices (on the page 3).

* In Eq. (96), what is the X appearing in e_1? I think it should be f of Eq. (95). Then, what is the X appearing in the paragraph under Eq. (98)?

-->>Thanks for finding the error. We corrected it.

* The manuscript is plagued with typos (mainly misspelled words). The authors should re-read it and correct them.​

-->>Errors have been corrected as much as possible.​ Thanks again.

Reviewer 2 Report

I have read the paper thoroughly. The author evolves the spinor formalism in General Relativity (GR), the material is presented clearly and logically, the results are correct. However, I am of the opinion that the work has a shortcoming – there is no part or section that indicates applicability of the results obtained to solve the theoretical problems of physics associated with gravity and cosmology. As the journal Universe mainly presents studies in theoretical physics, such a part
is essential. Otherwise, the author’s results can be considered as elaboration of the mathematical methods only. Such a part may be not very large but without its addition I can’t recommend the publication. I think that after inclusion of such a part into the paper considered, it may be published in the journal Universe.

Author Response

Answers to Referee 2

I have read the paper thoroughly. The author evolves the spinor formalism in General Relativity (GR), the material is presented clearly and logically, the results are correct. However, I am of the opinion that the work has a shortcoming – there is no part or section that indicates applicability of the results obtained to solve the theoretical problems of physics associated with gravity and cosmology. As the journal Universe mainly presents studies in theoretical physics, such a part 

is essential. Otherwise, the author’s results can be considered as elaboration of the mathematical methods only. Such a part may be not very large but without its addition I can’t recommend the publication. I think that after inclusion of such a part into the paper considered, it may be published in the journal Universe.​

-->>We further proposed the application in modified gravity and quantum gravity.  

We added a section (subsection VI-D) in page 22.

And the conclusions were also modified to contain more physical interpretations.​

Round 2

Reviewer 1 Report

I think that the new version of the paper has been improved with the new additions. Moreover, all the points raised in my report were addressed. Therefore, I recommend this paper for publication.

Reviewer 2 Report

I carefully read the revised version of the article. I see that the authors fully took into account my remark. I agree with the publication of this work in its current form